# Differences and agreement between two portable hand-held spirometers across diverse community-based populations in the Prospective Urban Rural Epidemiology (PURE) study

**MyLinh Duong**[1]*, **Sumathy Rangarajan**[1], **Michele Zaman**[1], **Nafiza Mat Nasir**[2], **Pamela Seron**[3], **Karen Yeates**[4], **Afzalhussein M. Yusufali**[5], **Rasha Khatib**[6], **Lap Ah Tse**[7], **Chuangshi Wang**[8], **Andreas Wielgosz**[9], **Koon Teo**[1], **Rajesh Kumar**[10], **Alvaro Avezum**[11], **Rosnah Ismail**[12], **Burcu Tumerdem çalık**[13], **Soumya Gopakumar**[14], **Omar Rahman**[15], **Katarzyna Zatońska**[16], **Annika Rosengren**[17], **Johanna Otero**[18], **Roya Kelishadi**[19], **Rafael Diaz**[20], **Thandi Puoane**[21], **Salim Yusuf**[1]

1 Department of Medicine, Population Health Research Institute, McMaster University and Hamilton Health Sciences, Hamilton, Ontario, Canada, 2 Faculty of Medicine, Universiti Teknologi MARA, Sungai Buloh Campus, Selangor, Malaysia, 3 Facultad de Medicina, Universidad de La Frontera, Temuco, Chile, 4 Pamoja Tunaweza Research Centre, Moshi, Tanzania, 5 Dubai Medical University, Hatta Hospital, Dubai Health Authority, Dubai, United Arab Emirates, 6 Advocate Aurora Research Institute, Milwaukee, IL, United States of America, 7 JC School of Public Health and Primary Care, The Chinese University of Hong Kong, Shatin, New Territories, Hong Kong, 8 Medical Research & Biometrics Center, National Center for Cardiovascular Diseases, Fuwai Hospital, Chinese Academy of Medical, Beijing, China, 9 University of Ottawa Department of Medicine, Ottawa, Ontario, Canada, 10 State Health System Resource Center, Punjab, India, 11 International Research Center, Hospital Alemão Oswaldo Cruz, São Paulo, SP, Brazil, 12 Community Health Department, Faculty of Medicine, Universiti Kebangsaan Malaysia, Kuala Lumpur, Malaysia, 13 Faculty of Health Sciences, Department of Health Management, Marmara University, Istanbul, Turkey, 14 Health Action by People and Government Medical College, Thiruvananthapuram, Kerala, India, 15 University of Liberal Arts Bangladesh, Dhaka, Bangladesh, 16 Wroclaw Medical University Bujwida Wroclaw, Poland, EU, 17 University of Gothenburg, Gothenburg, Sweden, 18 Instituto Masira, Universidad de Santander (UDES), Bucaramanga, Colombia, 19 Cardiovascular Research Institute, Chamran Hospital, Isfahan, Iran, 20 Estudios Clinicos Latinoamerica ECLA Rosario, Santa Fe, Argentina, 21 University of the Western Cape, School of Public Health, Cape Town, South Africa

* duongmy@mcmaster.ca

## Abstract

### Introduction

Portable spirometers are commonly used in longitudinal epidemiological studies to measure and track the forced expiratory volume in first second ($FEV_1$) and forced vital capacity (FVC). During the course of the study, it may be necessary to replace spirometers with a different model. This raise questions regarding the comparability of measurements from different devices. We examined the correlation, mean differences and agreement between two different spirometers, across diverse populations and different participant characteristics.

### Methods

From June 2015 to Jan 2018, a total of 4,603 adults were enrolled from 628 communities in 18 countries and 7 regions of the world. Each participant performed concurrent

**Data Availability Statement:** The Population Health Research Institute (PHRI) is the sponsor of

this STUDY. The PHRI believes the dissemination of research results is vital and sharing of data is important. PHRI prioritizes access to data to researchers who have worked on the PURE study for a significant duration, have played substantial roles, and have participated in raising the funds to conduct the study. Data will be disclosed upon request and approval of the proposed use of the data by a PURE Review Committee. Specific collaborative projects can be developed with groups with similar data for joint analyses. The underlying data for this clinical study contains personal information and personal health information of participants who were involved, which is protected under Canada's privacy laws, HIPPA (US) and GDPR, amongst other international laws governing privacy. Consent for public disclosure of this information was not obtained and could pose a threat to confidentiality and violate privacy laws. PHRI has no objection in sharing the information under confidentiality and with appropriate data protection and privacy, including to the journal statisticians in a timely manner, for verification or validation of the analyses in the paper upon request. As per the Canadian funding body guidelines https://cihr-irsc. gc.ca/e/29072.html, (referenced by PLOS), Element 8: "there should be strict limits on access to data and secure procedures for data linkage, subject to data-sharing agreements". PHRI follows this procedure and does not share or link data from clinical studies publicly where such data is or contains personal health information. Requests for access to data may be sent to PURE Publications Committee and the PHRI Contracts phri. contracts@phri.ca.

**Funding:** The funding for the main study of PURE is provided in the accompanying appendix. The current substudy is not funded. The authors received no specific funding for this work. The funders of the main study of PURE had no role in study design, data collection and analysis, decision to publish, or preparation of the manuscript.

**Competing interests:** All authors have declared that they have no competing interests.

measurements from the MicroGP and EasyOne spirometer. Measurements were compared by the intra-class correlation coefficient (ICC) and Bland-Altman method.

## Results

Approximately 65% of the participants achieved clinically acceptable quality measurements. Overall correlations between paired $FEV_1$ (ICC 0.88 [95% CI 0.87, 0.88]) and FVC (ICC 0.84 [0.83, 0.85]) were high. Mean differences between paired $FEV_1$ (-0.038 L [-0.053, -0.023]) and FVC (0.033 L [0.012, 0.054]) were small. The 95% limits of agreement were wide but unbiased (FEV1 984, -1060; FVC 1460, -1394). Similar findings were observed across regions. The source of variation between spirometers was mainly at the participant level. Older age, higher body mass index, tobacco smoking and known COPD/asthma did not adversely impact on the inter-device variability. Furthermore, there were small and acceptable mean differences between paired $FEV_1$ and FVC z-scores using the Global Lung Initiative normative values, suggesting minimal impact on lung function interpretation.

## Conclusions

In this multicenter, diverse community-based cohort study, measurements from two portable spirometers provided good correlation, small and unbiased differences between measurements. These data support their interchangeable use across diverse populations to provide accurate trends in serial lung function measurements in epidemiological studies.

## Introduction

Lung function assessments are now more accessible with the wide adoption of handheld portable spirometers in the community and ambulatory care setting. These devices are easy to operate and many have inbuilt quality check software to enable high quality measurements. They are also commonly employed in research studies to provide rapid and reliable lung function measurements and tracking of lung function over time [1]. However, in large multicenter trials, it is common to have different portable spirometers across different study sites depending on the local availability of these devices; and it is often necessary to replace older devices with newer models over time [2]. This raise questions regarding the reliability and agreement between measurements obtained from different devices. Therefore, it is important to ascertain the reliability, differences and agreement between different spirometers; and identify factors that may contribute to the variability between spirometers.

To date, there have been few small studies, which examined the variability between different portable spirometers [2–11]. Many were conducted in highly selected healthy young individuals in laboratory setting. Only a few were conducted in the community but limited to one population (generally from Europe or North America). It is unclear whether these findings can be generalized to other populations with different anthropometrics, demographics and underlying disease prevalence. Furthermore, not much is known on the source of variability between spirometers.

The Prospective Urban Rural Epidemiology (PURE) study is an international prospective cohort study, comprising of adults recruited from urban and rural communities from high, middle and low-income countries. Baseline spirometry data was collected with a handheld portable turbine spirometer without flow volume loops (FVL). In the course of cohort follow-

up, a new portable ultrasonic spirometer was introduced, which provided FVL. In the present study, we examined the correlation, agreement and mean difference between measurements from the old and new spirometer, in an unselected sub-sample. We also assessed whether the correlation and agreement between spirometers may differ across diverse populations from different socioeconomic and geographic regions. Lastly, we examined the impact of utilizing two different spirometers on the interpretation of spirometry measurements, using the Global Lung Initiative (GLI) normative values. Our findings will address some of the challenges associated with the widespread use of portable spirometers and their role in providing access to lung function measurements in the community. This information will facilitate correct interpretation of data and offer insight into how best to address the variability between spirometers.

## Methods

The PURE study began recruitment in 2004 of community-based adults aged 35 to 70 years old; from 628 urban and rural communities across 18 high-, middle- and low-income countries. The study design and methodology have been described elsewhere [12]. In brief, standardized approaches were used for the enumeration of households, identification of participants, recruitment and data collection. As it was not feasible to collect data from a representative sample of each country, the sampling method used for each country aimed to reduce participation bias based on local risk factors and disease prevalence. Baseline data were collected between 2004–2009 and follow-up occurred every 3 years. The study is coordinated by the Population Health Research Institute, Hamilton Health Sciences, McMaster University (Hamilton, ON, Canada). Ethics approval was provided by the Hamilton Health Sciences Research Ethics Board and the research ethics committees of the other participating centers (Appendix I in S1 File). All participants provided written informed consent to participate in the study.

Baseline spirometry was measured with the MicroGP spirometer (MicroMedical, Chatham. IL, USA), without FVL, following the 2005 American Thoracic Society/European Respiratory Society (ATS/ERS) spirometry standardization guidelines [13]. The MicroGP spirometer contains a turbine, which generates rotational flow during the spirometry maneuver. The rotation of the low-inertia vane is converted into electrical impulses by means of an infrared light-emitting diode and a photodiode sensor. A microprocessor within the device converts the electrical pulses into spirometry measurements, which are displayed digitally. According to the manufacturer, the microGP has an accuracy of ±2%. In 2015, the EasyOne (Ndd, Medical Technologies, Inc., Switzerland) ultrasonic spirometer was introduced, which provided automated quality checks, messaging, quality grades and FVL. The quality grades after each test session provided by the EasyOne include: (1) Grades A or B for three acceptable efforts and <100 ml (Grade A) or <150 ml (Grade B) variability between the two highest $FEV_1$ and FVC; (2) Grade C for two or more acceptable efforts and <200 ml variability; (3) Grade D for one acceptable effort or highly variable efforts $> = 200$ ml; and (4) Grade F for no acceptable efforts. The EasyOne spirometer uses an ultrasonic sensor to measure airflow. It has no moving parts and its accuracy is not dependent on mechanical function or the measurement of pressure or volume displacement. Accordingly, the manufacturer information report an accuracy <3%, which is maintained throughout its operational life and not needing regular calibration.

All study visits were conducted in dedicated research clinics in the community for all sites and countries. Participants were coached by a trained staff, prior to performing pre-bronchodilator forced inspiratory and expiratory manoeuvers (up to six attempts). All tests were performed in a standing position with participants' back straight and wearing a nose-clip. With

the introduction of the EasyOne spirometer, each center enrolled the first five consecutive participants from each community into the present substudy. Each participant provided spirometry measurements using the two devices in a random order within 3 hours supervised by the same research staff. The order of spirometer measurements was randomly generated by the coordinating site and issued to the center prior to the day of testing. Spirometers were calibrated monthly (or as needed in extreme weather or handling) using a 3L syringe to ensure an accuracy <105 ml or 3.5%.

## Statistical analysis

Means and frequency statistics were used to describe the data. The highest $FEV_1$ and FVC from each spirometer were analyzed. The assumption of normality and constant variance of the $FEV_1$ and FVC were assessed by visual inspection of histograms and plots of residuals against fitted values. The correlation and agreement between spirometers were assessed with scatterplots, intra-class correlation coefficients (ICC) and Bland-Altman plots [14]. Mean differences between paired $FEV_1$ and paired FVC were calculated as absolute (EasyOne–MicroGP) and relative ([EasyOne-MicroGP]/ average]*100) differences between spirometers. The random-intercept multilevel 'null' model was used to estimate the source (region, country, center and participant level) of variation between spirometers. Stratified analyses by region, sex, age, body mass index (BMI), smoking status, known COPD or asthma, education level and quality grades were performed to explore the effect of each factor on the reliability and agreement between spirometers. Countries were classified into seven regions according to geographic location and socioeconomic level (by the World Bank classification) [15]. To examine the impact on interpretation, the GLI normative values were used to transform $FEV_1$ and FVC into z-scores prior to Bland-Altman analysis [16]. We used the ATS/ERS recommendation for between-effort repeatability within test session of <150 ml to assess whether mean differences between spirometers met the criterion for within test reproducibility [17]. Similarly, a difference in z-score <0.5 SD was regarded as not meaningful difference between age, sex, height and ethnicity GLI adjusted values [18]. All analyses were performed using SAS version 9.4 (The SAS Institute, Cary, NC, USA) and STATA 15 (StataCorp LLC, Texas, USA).

## Results

A total of 4,603 participants from 628 communities in 18 countries across 7 regions completed measurements from the two spirometers. Baseline characteristics of included participants are shown in Table 1. Similar to the larger PURE study (Appendix II in S1 File), there were more females and individuals between the ages of 50–65 years. The overall proportion of participants meeting quality grades A, B or C on the EasyOne device was 65%, which is similar to the larger PURE study. There was a trend for higher prevalence of comorbidities including COPD/asthma and cardiac diseases; and lower education level in the substudy.

The correlations, mean differences and limits of agreement (LoA) between paired $FEV_1$ and FVC by region are shown in Table 2. Overall, paired $FEV_1$ and FVC between spirometers were highly correlated (Fig 1). The overall mean differences between spirometers, whether in absolute volume or as a percentage of mean $FEV_1$ or FVC were small and within acceptable limits of between-effort reproducibility (Fig 2). The 95% LoA between paired measurements were wide and showed no association with the size of $FEV_1$ or FVC. Correlations between paired $FEV_1$ and FVC were similarly high across regions except for South Asia, where there were low to moderate strength of correlation (Table 2). For South America and the Middle East, the correlation between paired FVC were lower than $FEV_1$. Across regions, the mean differences between paired $FEV_1$ were small (range from absolute -83ml [relative difference -4%]

**Table 1. Baseline characteristics by region.**

|  | Overall | S Asia | China | S East Asia | Africa | Middle East | S America | N Am/Eur |
|---|---|---|---|---|---|---|---|---|
| N (total) | 4603 | 566 (12.3) | 631 (13.7) | 191 (4.1) | 368 (8) | 433 (9.4) | 1,578 (34.3) | 836 (18.2) |
| Females | 2,840 (61.7) | 335 (59.2) | 351 (55.6) | 101 (52.9) | 257 (69.8) | 306 (70.7) | 1,044 (66.2) | 446 (53.3) |
| Urban | 2,363 (51.4) | 332 (58.8) | 119 (18.9) | 54 (28.3) | 165 (44.8) | 243 (56.1) | 854 (54.1) | 596 (71.3) |
| Age, years | 49.6 ± 9.2 | 46.2 ± 9.2 | 50.0 ± 8.5 | 52.9 ± 8.9 | 49.2 ± 9.2 | 47.0 ± 9.2 | 50.3 ± 9.1 | 52.2 ± 9.0 |
| Weight, (kg) | 70.2 ± 15.9 | 61.2 ± 12.6 | 64.7 ± 11.6 | 64.9 ± 13.5 | 67.3 ± 18.4 | 78.8 ± 15.9 | 70.3 ± 14.5 | 78.4 ± 16.8 |
| BMI (kg/m$^2$) | 27 ± 5.5 | 24.8 ± 4.8 | 25.1 ± 3.8 | 26.1 ± 4.9 | 26.5 ± 7.4 | 30.6 ± 5.9 | 27.7 ± 5.1 | 28.1 ± 5.5 |
| Height (cm) | 160.6 (9.6) | 156.9 ± 8.9 | 160.3 ± 8.4 | 157.5 ± 9.2 | 159.8 ± 8.5 | 160.5 ± 8.7 | 159.3 ± 9.2 | 166.8 ± 9.9 |
| COPD/asthma | 293 (6.4) | 19 (3.4%) | 36 (5.7) | 11 (5.8%) | 11 (3.0) | 36 (8.3) | 72 (4.6) | 108 (12.9) |
| Cardiac disease | 253 (5.5) | 33 (5.8) | 55 (8.7) | 5 (2.6) | 11 (3) | 31 (7.2) | 66 (4.2) | 52 (6.2) |
| Strokes | 110 (2.4) | 16 (2.8) | 33 (5.2) | 4 (2.1) | 3 (0.8) | 6 (1.4) | 30 (1.9) | 18 (2.1) |
| Smoking status |  |  |  |  |  |  |  |  |
| • current | 628 (13.7) | 88 (15.6) | 119 (19.1) | 25 (13.1) | 107 (29.1) | 40 (9.2) | 155 (9.8) | 94 (11.3) |
| • ex-smokers | 954 (20.8) | 27 (4.8) | 145 (23.2) | 27 (14.1) | 20 (5.4) | 26 (6) | 429 (27.2) | 280 (33.5) |
| • never | 3,013 (65.5) | 451 (79.6) | 360 (57.7) | 139 (72.8) | 241 (65.5) | 367 (84.8) | 994 (63) | 461 (55.2) |
| Education |  |  |  |  |  |  |  |  |
| primary/below | 2,368 (51.7) | 314 (56) | 322 (51.4) | 87 (45.5) | 280 (77.8) | 197 (45.5) | 1027 (65.2) | 141 (16.9) |
| secondary/above | 2,213 (48.3) | 247 (44) | 305 (48.6) | 104 (54.5) | 80 (22.2) | 236 (54.5) | 549 (34.8) | 692 (83.1) |
| FEV$_1$, L, Micro | 2.21 ± 0.8 | 1.76 ± 0.72 | 2.10 ± 0.65 | 1.90 ± 0.72 | 1.87 ± 0.67 | 2.26 ± 0.78 | 2.24 ± 0.79 | 2.74 ± 0.77 |
| z score, Micro | -1.06 ± 1.74 | -1.52 ± 1.56 | -1.46 ± 1.62 | -1.52 ± 1.56 | -1.05 ± 1.67 | -1.36 ± 1.58 | -0.82 ± 1.86 | -0.42 ± 1.30 |
| FEV$_1$, L Easy | 2.18 ± 0.7 | 1.71 ± 0.56 | 2.11 ± 0.65 | 1.84 ± 0.60 | 1.80 ± 0.61 | 2.23 ± 0.69 | 2.20 ± 0.72 | 2.68 ± 0.74 |
| z score, Easy | -1.18 ± 1.45 | -1.54 ± 1.14 | -1.31 ± 1.59 | -1.54 ± 1.14 | -1.18 ± 1.48 | -1.58 ± 1.32 | -1.08 ± 1.49 | 0.53 ± 1.16 |
| FVC, L, Micro | 2.71 ± 1.1 | 1.98 ± 0.95 | 2.65 ± 0.85 | 2.22 ± 0.79 | 2.38 ± 0.96 | 2.90 ± 1.13 | 2.69 ± 1.02 | 3.46 ± 1.01 |
| z score, Micro | -1.21 ± 1.85 | -1.68 ± 1.63 | -1.52 ± 1.83 | -1.69 ± 1.63 | -1.04 ± 1.80 | -1.32 ± 1.94 | -1.22 ± 1.89 | -0.38 ± 1.27 |
| FVC, L Easy | 2.81 ± 0.9 | 2.17 ± 0.70 | 2.74 ± 0.81 | 2.29 ± 0.70 | 2.41 ± 0.73 | 2.73 ± 0.83 | 2.83 ± 0.85 | 3.44 ± 0.92 |
| z score Easy | -1.13 ± 1.54 | -1.79 ± (1.25 | -1.35 ± 1.79 | -1.79 ± 1.24 | -0.92 ± 1.48 | -1.81 ± 1.31 | -0.91 ± 1.47 | -0.49 ± 1.08 |
| Grades A-B | 2171 (47.2) | 223 (39.5) | 392 (62.1) | 92 (48.1) | 152 (41.3) | 214 (49.4) | 548 (34.7) | 551 (65.9) |
| Grade C | 783 (17) | 92 (16.3) | 100 (15.8) | 38 (20) | 71 (19.3) | 60 (13.9) | 309 (19.6) | 113 (13.5) |
| Grade D | 1170 (25.4) | 147 (26) | 123 (19.5) | 43 (22.5) | 95 (25.8) | 102 (23.6) | 534 (33.9) | 126 (15.1) |
| Grade F | 457 (9.9) | 96 (17) | 14 (2.2) | 18 (9.4) | 50 (13.6) | 56 (12.9) | 185 (11.7) | 38 (4.5) |

Variables are presented as means ±SD for continuous data and absolute numbers (% of total in each region/ column). Abbreviations: BMI = body mass index calculated as weight divided by height squared; COPD (chronic obstructive pulmonary disease)/asthma, CHF (congestive heart failure) and strokes were self-reported; FEV$_1$ = forced expiratory volume in the first second measured in liters (L); FVC = forced vital capacity in liters (L); z-scores were estimated using the Global Lung Function Initiative normative values appropriate for age, sex, height and ethnicity; Micro = microGP spirometer; Easy = EasyOne spirometer. For regions S = South; N Am/ Eur = North America/Europe. The grades were quality grades using ATS guideline provided by the EasyOne spirometer.

to 49ml [2.5%]) and showed no consistent bias across regions. The mean differences between paired FVC were larger, particularly for the Middle East (-203 ml [-6%]) and South America (141 ml [6.3%]) and again showed no consistent bias across regions. The 95% LoA were wide for both FEV$_1$ and FVC; suggesting large variation in agreement between spirometers across regions.

To understand the source of variation between spirometers, the ICC and variance components between spirometers were assessed at the region, country, center and participant levels (Table 3). The highest ICC between paired FEV$_1$ and FVC were observed at the participant level, indicating the measurements between spirometers were highly correlated within individuals. This correspond to the largest variance component, suggesting that participant factors contributed significantly to the variation between spirometers. The correlation and variance between spirometers at the region, country and center levels were substantially less, suggesting

**Table 2. Correlations, mean differences and agreement between spirometers by region.**

| | OVERALL | S Asia | China | SE Asia | Africa | Middle East | S America | N Am / Eur |
|---|---|---|---|---|---|---|---|---|
| **N** | 4603 | 566 | 631 | 191 | 368 | 433 | 1578 | 836 |
| **CORRELATIONS (ICC, 95%CI)** | | | | | | | | |
| **FEV$_1$** | **0.88** (0.87, 0.88) | **0.67** (0.61, 0.72) | **0.93** (0.91, 0.94) | **0.89** (0.85, 0.92) | **0.83** (0.79, 0.86) | **0.83** (0.80, 0.86) | **0.82** (0.80, 0.94) | **0.94** (0.93, 0.95) |
| **FVC** | **0.84** (0.83, 0.85) | **0.53** (0.45, 0.61) | **0.92** (0.91, 0.93) | **0.88** (0.85, 0.91) | **0.78** (0.73, 0.82) | **0.71** (0.64, 0.76) | **0.74** (0.71, 0.77) | **0.94** (0.93, 0.95) |
| **BLAND-ALTMAN ANALYSIS- mean differences (95% CI) and 95% limits of agreement (95% CI)** | | | | | | | | |
| **FEV$_1$, ml** | -38 (-53, -23) | -9 (-64, 47) | **49** (22, 76) | -3 (-47, 41) | -37 (-88, 14) | -83 (-135, -31) | -74 (-104, -45) | -42 (-68, -17) |
| Upper LoA | 984 (958, 1010) | 1300 (1205, 1394) | 725 (678, 771) | 1194 (1119, 1269) | 938 (851, 1026) | 984 (895, 1072) | 1082 (1032, 1132) | 696 (653, 740) |
| Lower LoA | -1060 (-1085, -1034) | -1317 (-1412, -1222) | -626 (-673, -580) | -1200 (-1275, -1125) | -1012 (-1099, -924) | -1150 (-1239, -1062) | -1231 (-1281, 1181) | -781 (-825, -737) |
| **FVC, ml** | **33** (12, 54) | **38** (-40, 117) | **51** (18, 84) | **19** (-42, 79) | **60** (-10, 131) | **-203** (-284, -122) | **141** (99, 182) | **-59** (-89, -30) |
| Upper LoA | 1460 (1424, 1496) | 1851 (1717, 1984) | 887 (830, 944) | 1655 (1551, 1759) | 1405 (1285, 1526) | 1476 (1337, 1615) | 1781 (1712, 1852) | 797 (746, 847) |
| Lower LoA | -1394 (-1430, -1358) | -1774 (-1908, -1640) | -785 (-842, -728) | -1618 (-1723, -1514) | -1285 (-1405, -1164) | -1882 (-2021, -1743) | -1500 (-1571, 1429) | -915 (-966, -865) |
| **FEV, %** | **-1.5%** (-2.3, -0.8) | **-1%** (-2.1, 4.3) | **2.5%** (1.1, 4) | **1.2%** (1.3, 3.7) | **-2.2%** (-4.9, 0.4) | **-4%** (-6.5, -1.6) | **-3.7%** (-5.2, -2.3) | **1.3%** (-2.5, -0.1) |
| Upper LoA | 51 (50,52) | 76 (70, 81) | 39 (37, 42) | 69 (65, 73) | 48 (44, 53) | 47 (43, 51) | 54 (51, 56) | 32 (30, 34) |
| Lower LoA | -54 (-55, -53) | -73 (-79, -68) | -34 (-37, -32) | -67 (-71, -63) | -53 (-57, -48) | -55 (-59, -51) | -61 (-64, -98) | -34 (-36, -34) |
| **FVC, %** | **2.3%** (1.5, 3.1) | **-3.3%** (-0.1, 6.7) | **2.3%** (0.9, 3.7) | **2.2%** (-0.4, 4.9) | **3.6%** (0.7, 6.5) | **-6%** (-9, -3) | **6.3%** (5, 8) | **-1.6%** (-2.4, -0.7) |
| Upper LoA | 56 (55, 58) | 83 (77, 89) | 37 (35, 40) | 74 (69, 78) | 60 (55, 67) | 54 (49, 59) | 65 (62, 67) | 24 (22, 25) |
| Lower LoA | -52 (-53, -50) | -76 (-82, -70) | -33 (-35, -30) | -69 (-74, -65) | -52 (-57, -47) | -66 (-71, -61) | -52 (-55, -50) | -27 (-28, -25) |

N = number of included participants within each region. The mean absolute difference (EasyOne value minus microGP value) or mean relative difference (EasyOne minus microGP)/average)*100) between spirometers are provided with 95% CI. The 95% upper and lower limits of agreement (LoA) and their 95% CI are provided. For regions S = South; SE = South East; N Am/Eur = North America/Europe.

these levels contribute substantially less to the variation between spirometers. Furthermore, the increase in size of the ICC and variance component from the region to country and center levels were not dramatic, compared to the large increase from center to participant levels. This further highlights the importance of participant factors in contributing to the variation between spirometers.

To explore the participant factors that may contribute to the variation between spirometers, we examined the baseline characteristics of participants, whose inter-device difference were within and outside the 95% LoA for the overall population (Appendix III in S1 File). The distribution in age, body mass index and sex were similar between these 2 groups. Furthermore, COPD/asthma, cardiac disease, strokes and tobacco smoking did not adversely impact the agreement between spirometers. However, there were higher percentages of lower quality grade spirometry and lower education level in those outside the 95% LoA. Separate stratified analyses were conducted to further explore the effects of sex, age, BMI, smoking status, known COPD or asthma, education and quality grades on spirometer variability (Table 4, Appendix IV in S1 File). The correlation between paired FEV$_1$ and FVC were generally high and similar across strata. The mean differences between spirometers were small with minimal variation

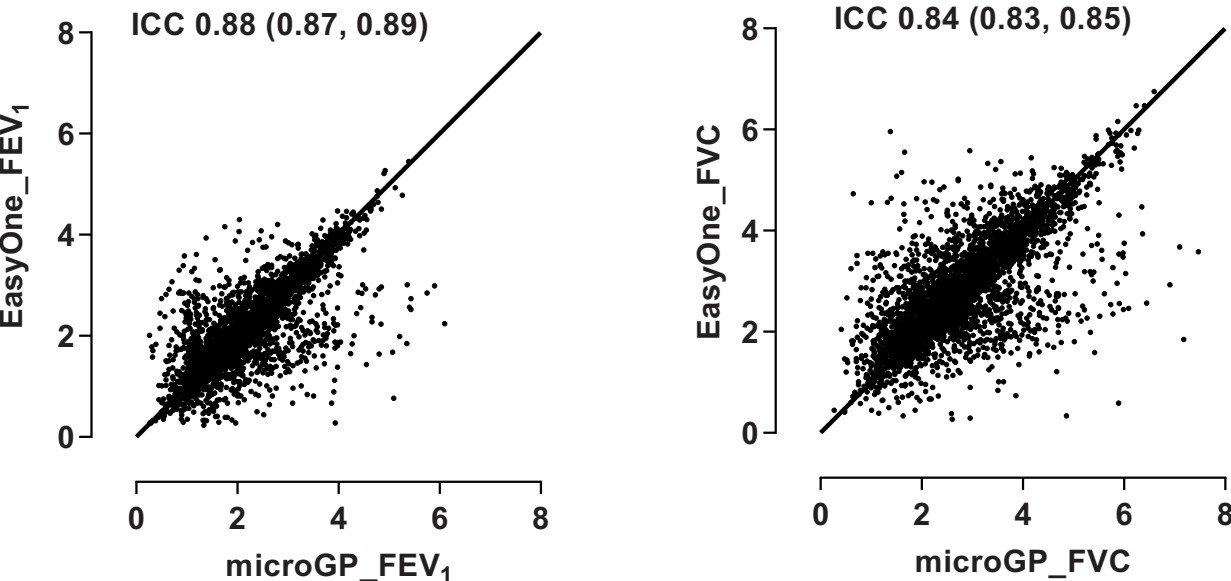

**Fig 1. Overall correlation between paired FEV1 and FVC from the microGP and EasyOne spirometers.** ICC = intraclass correlation coefficient and 95% CI for paired FEV1 (L) and FVC (L) measured within 3 hours and conducted in a random order. All measurements were supervised by the same trained study coordinator. The Line of identity is provide representing a perfect correlation between paired measurements.

across strata, even for the lower quality grades. However, there were lower correlation, larger variability and larger LoA between spirometers among those with lower education level and lower quality grades.

Similar Bland-Altman analyses were conducted on the $FEV_1$ and FVC z-scores using age, sex, height and ethnic appropriate GLI normative values (Table 5). Mean differences between paired $FEV_1$ and FVC z-scores from the two spirometers were small and less than 0.5 SD for the overall substudy and across regions.

## Discussion

In this large international multi-center community-based sub-study, we examined the correlation, mean differences and agreement between measurements from two commonly used portable spirometers used in the community and field studies; and how they may vary across diverse populations. We found an average of 65% of quality grades A, B and C, which are clinically acceptable efforts. The overall correlation between paired $FEV_1$ and paired FVC between spirometers were high. The overall mean differences between measurements were small and within acceptable limits of between-effort reproducibility. There were moderate to high correlations between spirometers across diverse populations from different geographic and socioeconomic regions. Mean differences between paired $FEV_1$ were uniformly small across regions, while larger differences between paired FVC were observed. In both cases, there was no systematic bias observed across region. The main source of variation between spirometers was at the participant level, with much less variation observed among regions, countries and centers. Exploratory analyses of participant factors identified low education level and poor quality grade efforts were associated with higher variability between spirometers.

As portable spirometers become widely adopted and used in the community, more information on their quality of measurements, reliability, biases and agreement are needed, which will enable correct interpretation and comparison of lung function data across spirometers.

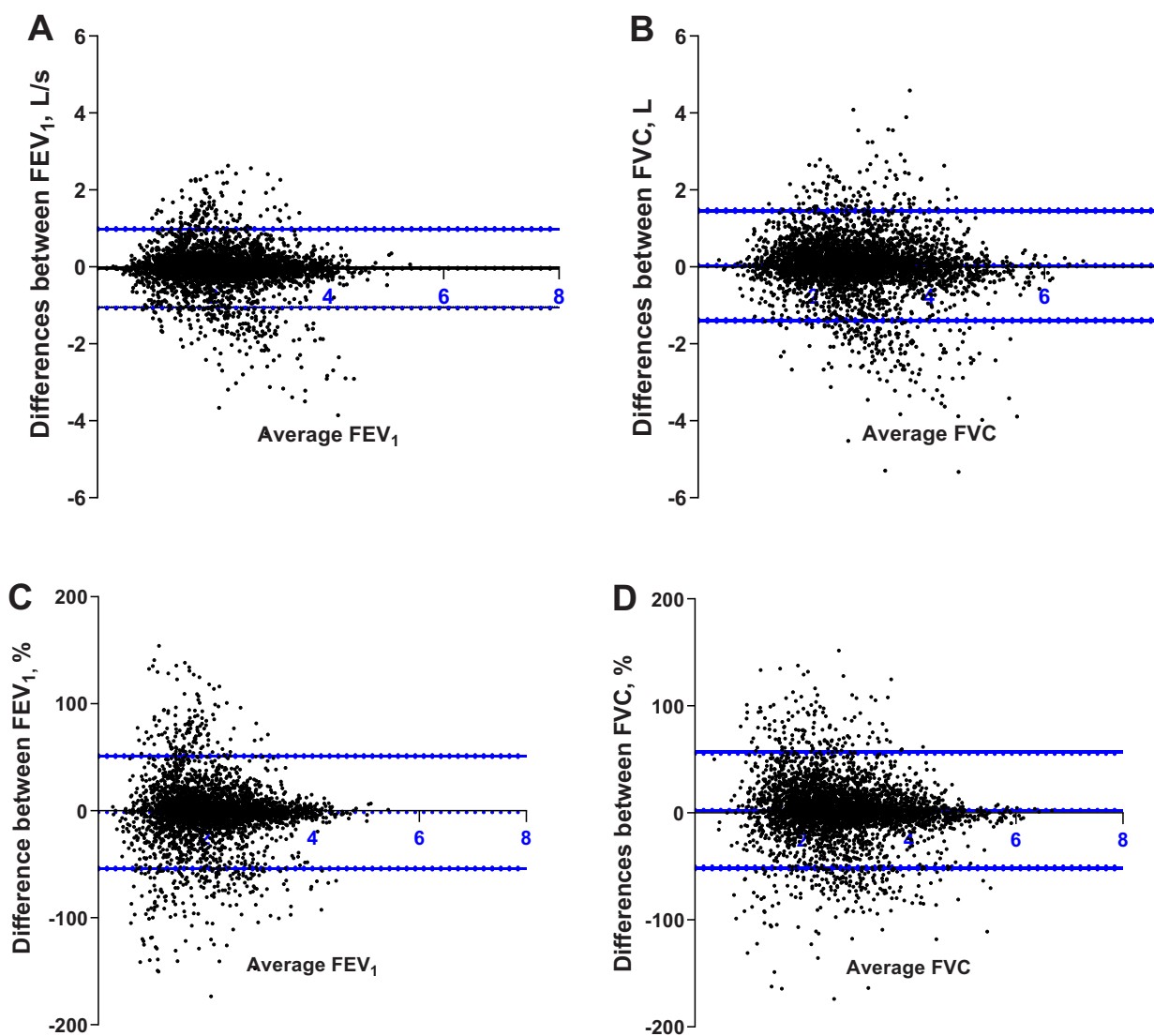

**Fig 2. Bland-Altman plots for paired FEV1 and FVC measured by the microGP abd EasyOne spirometers.** Differences between paired FEV1 and FVC were calculated as the absolute mean difference (EasyOne minus microGP) in Panels And B; or the relative mean difference (EasyOne minus microGP)/average * 100) in Panels C and D; plotted against the average ((EasyOne+microGP) 2) on the x-axis. The 95% Limits of Agreement (LoA) are provided (blur lines). The 95% CI for the mean differences and LoA are also provide (broken Line).

**Table 3. ICC and variance estimates between spirometer at the region, country, centers and individuals levels.**

| Levels | FEV$_1$ | | FVC | |
|---|---|---|---|---|
| | ICC (95% CI) | Variance (95%CI), Liters$^2$ | ICC (95% CI) | Variance (95%CI), Liters$^2$ |
| Region | 0.133 (0.039, 0.361) | 0.084 (0.023, 0.306) | 0.146 (0.042, 0.399) | 0.148 (0.039, 0.559) |
| Country within region | 0.168 (0.069, 0.356) | 0.023 (0.003, 0.155) | 0.202 (0.087, 0.402) | 0.056 (0.011, 0.291) |
| Centers within countries | 0.208 (0.104, 0.373) | 0.025 (0.009, 0.012) | 0.248 (0.129, 0.422) | 0.046 (0.022, 0.096) |
| Individuals within centers | 0.783 (0.743, 0.819) | 0.366 (0.348, 0.384) | 0.730 (0.673, 0.781) | 0.488 (0.463, 0.515) |
| Residual error | | 0.138 (0.132, 0.143) | | 0.273 (0.262, 0.284) |

ICC = intraclass coefficient.

**Table 4. Stratified analyses by demographic, anthropometric, clinical characteristics and quality grades.**

| SEX | | N | Correlation | mean Diff, ml | LoA, ml | mean Diff, % | LoA, % |
|---|---|---|---|---|---|---|---|
| FEV$_1$ | Females | 2,832 | 0.80 (0.78, 0.82) | -60 (-78, -42) | -1017, 897 | -3 (-4, -2) | -54, 49 |
| | Males | 1,756 | 0.87 (0.86, 0.89) | -2 (-28, 25) | -1117, 1114 | 0.7 (-0.6, 2) | -52, 54 |
| FVC | Females | 2,827 | 0.70 (0.68, 0.72) | -9 (-35, 19) | -1455, 1438 | 1.0 (-0.09, 2) | -56, 58 |
| | Males | 1,758 | 0.86 (0.84, 0.87) | 101 (68, 134) | -1284, 1486 | 4.4 (3.2, 5.6) | -45, 54 |
| AGE | | N | Correlation | mean Diff, ml | LoA, ml | mean Diff, % | LoA, % |
| FEV$_1$ | <50 y | 950 | 0.88 (0.86, 0.89) | -12 (-46, 22) | -1071, 1047 | 0.2 (-1.5, 1.9) | -51, 51 |
| | 50–65 y | 2494 | 0.87 (0.86, 0.88) | -31 (-52, -10) | -1058, 996 | -1.3 (-2.3, -0.2) | -53, 50 |
| | >65 y | 1147 | 0.87 (0.85, 0.88) | -74 (-103, -45) | -1049, 901 | -3.6 (-5, -2) | -58, 51 |
| FVC | <50 y | 793 | 0.84 (0.82, 0.86) | 56 (8, 105) | -1431, 1544 | 3.4 (1.7, 5) | -48, 55 |
| | 50–65 y | 2483 | 0.84 (0.82, 0.85) | 37 (9, 65) | -1363, 1437 | 2.3 (1.3, 3.4) | -50, 55 |
| | >65 y | 1,094 | 0.82 (0.80, 0.84) | 7 (-36, 49) | -1427, 1441 | 1.3 (-0.5, 3) | -58, 60 |
| BMI | | N | Correlation | mean Diff, ml | LoA, ml | mean Diff, % | LoA, % |
| FEV$_1$ | <18 | 100 | 0.80 (0.70, 0.86) | -119 (-246, 8.6) | -1374, 1137 | -8 (-15, -0.6) | -78, 63 |
| | 18–25 | 1588 | 0.88 (0.87, 0.89) | -20 (-45, 5) | -1024, 984 | -0.7 (-2, 0.6) | -53, 51 |
| | >25 | 2883 | 0.87 (0.86, 0.88) | -43 (-62, -24) | -1060, 975 | -1.6 (-2.6, -0.7) | -53, 50 |
| FVC | <18 | 99 | 0.71 (0.57, 0.80) | -98 (-291, 96) | -2000, 1805 | -2 (-9, 6) | -77, 75 |
| | 18–25 | 1577 | 0.85 (0.83, 0.86) | 76 (41, 111) | -1326, 1478 | 3.7 (2.4, 5) | -51, 58 |
| | >25 | 2877 | 0.84 (0.83, 0.85) | 17 (-9, 43) | -1397, 1431 | 1.7 (0.8, 2.7) | -51, 54 |
| SMOKING | | N | Correlation | mean Diff, ml | LoA, ml | mean Diff, % | LoA, % |
| FEV$_1$ | Never | 2,998 | 0.86 (0.85, 0.87) | -34 (-53, -14) | -1101, 1033 | -1.3 (-2.3, -0.3) | -56, 53 |
| | Ever | 1582 | 0.90 (0.89, 0.91) | -46 (-70, -23) | -979, 886 | -2 (-3, -0.8) | -50, 46 |
| FVC | Never | 2984 | 0.82 (0.81, 0.83) | 11 (-17, 38) | -1478, 1497 | 2 (0.6, 2.7) | -55, 59 |
| | Ever | 1576 | 0.86 (0.85, 0.88) | 75 (42, 108) | -1230, 1380 | 3 (2, 5) | -45, 52 |
| COPD/ ASTHMA | | N | Correlation | mean Diff, ml | LoA, ml | mean Diff, % | LoA, % |
| FEV$_1$ | No | 4295 | 0.87 (0.86, 0.88) | -38 (-53, -22) | -1073, 998 | -1.4 (-2.2, -0.6) | -54, 52 |
| | Yes | 293 | 0.93 (0.92, 0.95) | -41 (-88, 5) | -832, 749 | -3.4 (-5.8, -0.9) | -46, 39 |
| FVC | No | 4275 | 0.83 (0.82, 0.84) | 37 (15, 59) | -1405, 1480 | 2.5 (1.6, 3.3) | -52, 57 |
| | Yes | 293 | 0.90 (0.87, 0.92) | - 23 (-92, 46) | -1201, 1154 | -0.3 (-3, 2.5) | -48, 48 |
| EDUCATION | | N | Correlation | mean Diff, ml | LoA, ml | mean Diff, % | LoA, % |
| FEV$_1$ | Low | 2,362 | 0.82 (0.80, 0.83) | -68 (-92, -46) | -1189, 1052 | -3.3 (-4.4, -2.1) | -61, 54 |
| | high | 2,207 | 0.91 (0.90, 0.92) | -5 (-24, 14) | -904, 894 | 0.3 (-0.7, 1.2) | -46, 47 |
| FVC | low | 2,360 | 0.76 (0.74, 0.77) | 18 (-15, 51) | -1566, 1602 | 1.6 (0.3, 2.9) | -61, 64 |
| | high | 2,205 | 0.88 (0.87, 0.89) | 49 (22, 77) | -1230, 1329 | 2.8 (1.8, 3.8) | -43, 49 |
| QUALITY GRADES | | N | Correlation | mean Diff, ml | LoA, ml | mean Diff, % | LoA, % |
| FEV$_1$ | A | 1,555 | 0.88 (0.87, 0.89) | -39 (-58, -19) | -811, 734 | -1.2 (-2.2, -0.2) | -39, 37 |
| | B | 611 | 0.90 (0.88, 0.91) | -16 (-15, 45) | -729, 759 | 1.2 (-0.2, 2.7) | -36, 38 |
| | C | 779 | 0.81 (0.78, 0.83) | -17 (-49, 16) | -924, 890 | -0.2 (-1.8, 1.4) | -46, 45 |
| | D/F | 1,637 | 0.65 (0.62, 0.68) | -67 (-100, -35) | -1391, 1256 | -3.4 (-5.2, -1.7) | -73, 66 |
| FVC | A | 1,551 | 0.78 (0.76, 0.80) | -27 (-55, 2) | -1152, 1099 | -0.01 (-1, 1) | -42, 42 |
| | B | 611 | 0.84 (0.79, 0.85) | 45 (1, 89) | -1044, 1134 | 2.7 (1.1, 4.4) | -38, 44 |
| | C | 776 | 0.76 (0.62, 0.77) | 12 (-35, 59) | -1289, 1313 | 1.7 (-0.02, 3.4) | -46, 49 |
| | D/F | 1,624 | 0.61 (0.57, 0.64) | 95 (51, 140) | -1702, 1893 | 4.5 (2.8, 6.2) | -65, 74 |

Analyses were stratified by sex; age; body mass index (BMI); smoking (EVER included current and ex-smokers of tobacco products); known self-reported COPD or asthma; low education level = primary school and lower; high education = secondary school and higher; quality grades were provided by the EasyOne spirometer. The mean absolute difference (EasyOne minus -microGP value) or mean relative difference (EasyOne minus microGP/average)*100) between spirometers are provided with 95% CI. The 95% upper and lower limits of agreement (LoA) are provided. These data are also graphically represented in Appendix IV in S1 File.

**Table 5. Mean differences and agreement between Z-scores from the two spirometers for overall study population and by region.**

|  | OVERALL | S Asia | China | SE Asia | Africa | Middle East | S America | NAm/Eur |
|---|---|---|---|---|---|---|---|---|
| N | 4574 | 550 | 630 | 191 | 368 | 428 | 1571 | 836 |
| $FEV_1$ (95% LOA) | **-0.13** (2.7, -3.0) | **-0.088** (-3.6, 3.4) | **0.15** (-1.9, 2.2) | **-0.021** (-2.3, 2.3) | **-0.13** (-2.9, 2.6) | **-0.22** (-3.1, 2.7) | **-0.25** (-3.6, 3.1) | **-0.10** (-1.7, 1.4) |
| FVC (95% LOA) | **0.073** (-3.3, 3.4) | **0.055** (-4.6, 4.7) | **0.17** (-2.6, 2.9) | **-0.11** (-2.5, 2.2) | **0.12** (-3.0, 3.3) | **-0.49** (-4.2, 3.2) | **0.31** (-3.4, 4.0) | **-0.11** (-1.8, 1.6) |

$FEV_1$ = forced expiratory volume in the first second measured; FVC = forced vital capacity; LOA = 95% limits of agreement; z-scores were estimated using the Global Lung Function Initiative normative values appropriate for age, sex, height and ethnicity. Mean differences and LOA provided were calculated using Bland-Altman approach.

To date, most studies have compared different portable devices in highly selected healthy and mainly young non-smokers within a single population [2–11]. These studies have reported on high correlation and agreement between devices, which are likely to be inflated given the controlled setting under which the comparisons were made. The relatively small sample sizes and homogeneity of the population studied also limit the ability of prior studies to adequately address the source of variation between spirometers. In contrast, we examined two commonly used portable spirometers in large numbers of unselected individuals, from a wide range of urban and rural communities, and geographic regions. The measurements were collected outside of controlled laboratory setting, which can lend our findings more generalizable to a broader range of populations and settings.

Similar to other community-based studies, we found an average of 25 to 35% of suboptimal quality grade efforts [19]. Even with these data included, there were high correlations and small mean differences between paired $FEV_1$ across regions. For paired FVC, there was more variation in the correlation and mean differences between spirometers. However, for most regions, the mean differences between paired FVC still remained within the acceptable limits of between-effort reproducibility [17]. Furthermore, we observed no consistent bias between spirometers across regions suggesting the variation between devices was random in nature. We found the LoA were wide and variable across regions for both $FEV_1$ and FVC. This was expected as other studies have shown that the LoA will tend to increase with larger sample size and including wider range of data examined [20]. Also, in keeping with previous findings, we observed larger LoA between paired FVC than $FEV_1$ [2, 21].

To date, there has been very limited information on the source of variability between spirometers. The few studies that have examined the effect of age and sex on inter-device variability have reported on disparate findings [2, 7, 9]. These studies were generally small in sample size and included healthy volunteers across a limited age range. Our large sample size and diverse population enabled a robust analysis of the potential sources of variation between spirometers at the region, country, center and participant levels. We identified the largest source of variability was at the participant level, with much smaller contribution at the region, country or center levels. Importantly, participant factors such as older age, higher BMI, previous and current smoking and known COPD/asthma did not adversely affect the variation between spirometers. However, low education level and poor-quality grade efforts, were more likely to demonstrate lower correlation and larger variation between spirometers. Even in these subgroups, the mean differences between spirometers remained small and unbiased, suggesting sufficient precision and comparable estimates of group means across devices.

Our findings have a number of implications. First, we report on the robustness of the $FEV_1$ measurement, which was highly correlated, with small and unbiased differences between devices across diverse populations. The correlation and mean differences between paired FVC, however, were more variable but unbiased across regions. This suggests that a more

customized approach by region may be needed to adjust for the larger differences in the FVC between spirometers. Second, the LoA were wide, but random, suggesting considerable between-subject variability in agreement between devices. In this regard, it is important to differentiate the need for individual versus group level precision in estimating lung function for different types of studies. In population-based studies, where exclusion of participants is undesirable (since excluded participants may be systematically different from those included) this will inherently lead to larger inter-subject variability. Furthermore, the focus of population-based studies is mainly on the average differences in lung function between populations or the mean changes over time. In this context, it is more relevant to determine whether on average the recordings from different devices are well correlated, and collected without systematic bias. Therefore, the precision of group mean estimates to provide accurate trends is more important than the precision of individual measurements. By contrast, in clinical studies the within-subject variability may be more relevant in assessing changes in lung function within individuals or small groups in response to an intervention. Here the precision of individual measurements is likely to be more important. To that end, our findings suggest that the two different spirometers, on average, were highly correlated, and had sufficiently high precision in estimating the group means in the overall population and in key subgroups without bias. Furthermore, when the data were transformed using GLI normative values, we observed very small and acceptable differences in the mean z-scores across spirometers; suggesting limited impact on interpretation of the data. Lastly, we did not observe a large contribution to the variation between spirometers at the region, country or center levels, suggesting consistent execution of spirometry measurements across these levels. The main source of variation identified was at the participant level and may be related to factors such as low education level and poor quality spirometry efforts. To this end, while every reasonable effort should be made to increase the precision of individual lung function measurements; those that are beyond what is easily achievable, may not necessarily increase the power of the study but could lead to considerable increase in the complexity and cost of the study and therefore comprise study feasibility [22]. Moreover such methods may create biases (and distort results) particularly if such stringent criteria exclude participants with specific conditions or demographics that may influence lung function.

The strengths of our study include the large sample size, the diverse and unselected populations, which increases the generalizability of our findings. Measurements were taken in random order and supervised by the same-trained staff, and therefore minimize procedure-related variability. Furthermore, all spirographs available from the EasyOne were inspected and assessed by a staff respirologist to ensure agreement with the assessment. Limitations include the measurements of lung function without bronchodilation. The use of bronchodilation can help to reduce variable airway tone in asthmatic patients, which may contribute to the variation between spirometers. However, participants were not requested to withhold any medications prior to testing. Therefore, it is reasonable to assume, that those with chronic lung diseases including asthma would have taken their inhaler medications prior to spirometry assessments; and therefore are less likely to exhibit variable airway tone.

In conclusion, we found moderate to high correlation and small mean differences between paired $FEV_1$ and FVC between the MicroGP and EasyOne spirometers across diverse populations. The differences between paired measurements showed no consistent biases across regions. Our findings support the use of these two spirometers in large long-term studies to provide reliable and comparable measurements, with highly correlated and small unbiased differences between group means across diverse population.

## Supporting information

**S1 File.**
(DOC)

## Acknowledgments

We would like to acknowledge the assistance of the following members of our team who were involved in the collection, cleaning and validation of the spirometry data: Maha Mushtaha, Roxanna Solano, Justina Greene, Steven Chen and Alex Dragoman. We also would like to acknowledge the statistical help and assistance from Dr Shrinkant Bangdiwala and Ms Chinthanie Ramasundarahettige.

## Author Contributions

**Conceptualization:** MyLinh Duong, Salim Yusuf.

**Data curation:** MyLinh Duong, Sumathy Rangarajan, Nafiza Mat Nasir, Pamela Seron, Karen Yeates, Afzalhussein M. Yusufali, Rasha Khatib, Lap Ah Tse, Chuangshi Wang, Andreas Wielgosz, Koon Teo, Rajesh Kumar, Alvaro Avezum, Rosnah Ismail, Burcu Tumerdem çalık, Soumya Gopakumar, Omar Rahman, Katarzyna Zatońska, Annika Rosengren, Johanna Otero, Roya Kelishadi, Rafael Diaz, Thandi Puoane.

**Formal analysis:** MyLinh Duong, Michele Zaman, Salim Yusuf.

**Funding acquisition:** Sumathy Rangarajan, Salim Yusuf.

**Investigation:** MyLinh Duong, Michele Zaman, Andreas Wielgosz, Salim Yusuf.

**Methodology:** MyLinh Duong, Sumathy Rangarajan, Michele Zaman, Andreas Wielgosz, Koon Teo, Salim Yusuf.

**Project administration:** MyLinh Duong, Sumathy Rangarajan, Nafiza Mat Nasir, Salim Yusuf.

**Resources:** MyLinh Duong, Sumathy Rangarajan, Salim Yusuf.

**Supervision:** MyLinh Duong, Sumathy Rangarajan, Pamela Seron, Karen Yeates, Afzalhussein M. Yusufali, Rasha Khatib, Andreas Wielgosz, Koon Teo, Rajesh Kumar, Alvaro Avezum, Rosnah Ismail, Omar Rahman, Katarzyna Zatońska, Annika Rosengren, Johanna Otero, Roya Kelishadi, Rafael Diaz, Thandi Puoane, Salim Yusuf.

**Validation:** MyLinh Duong, Michele Zaman.

**Writing – original draft:** MyLinh Duong, Sumathy Rangarajan, Michele Zaman, Salim Yusuf.

**Writing – review & editing:** MyLinh Duong, Sumathy Rangarajan, Michele Zaman, Nafiza Mat Nasir, Pamela Seron, Karen Yeates, Afzalhussein M. Yusufali, Rasha Khatib, Lap Ah Tse, Chuangshi Wang, Andreas Wielgosz, Koon Teo, Rajesh Kumar, Alvaro Avezum, Rosnah Ismail, Burcu Tumerdem çalık, Soumya Gopakumar, Omar Rahman, Katarzyna Zatońska, Annika Rosengren, Johanna Otero, Roya Kelishadi, Rafael Diaz, Thandi Puoane, Salim Yusuf.

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
