## [Decision Letter · Decision Letter 0]

21 Jul 2021

PGPH-D-21-00083

Differences and agreement between two portable hand-held spirometers across diverse community-based populations in the Prospective Urban Rural Epidemiology (PURE) study.

Dear Dr. Duong,

Thank you for submitting your manuscript to PLOS Global Public Health. After careful consideration, we feel that it has merit but does not fully meet PLOS Global Public Health’s publication criteria as it currently stands. Therefore, we invite you to submit a revised version of the manuscript that addresses the points raised during the review process.

We look forward to receiving your revised manuscript.

Kind regards,

Andre F. S. Amaral, Ph.D.

Academic Editor

Journal Requirements:

Additional Editor Comments (if provided):

Reviewers' comments:

Reviewer #1: Thank you very much for the opportunity to review this manuscript. Duong and colleagues assess the comparison between two different spirometers in a very diverse population and in a larger study. This contributes good evidence on comparisons of spirometers especially in a real world setting as previous comparisons have been small and in a controlled setting.

Taking advantage of this large and diverse population, this reviewer feels the analyses can be expanded by comparing the two spirometers using the Global Lung Initiative reference equations. This would also help to provide very vital insight on how the GLI behave in this population as they have been indicated to behave differently.

This reviewer is also interested in the risk exposure-Lung function associations. Were these associations adjusted or crude? Were the lung function measurements mutually adjusted for? Since one measurement was at baseline and another at follow up? More specific details on this part of the analysis would benefit the reader. It would also be interesting to know whether these risk exposure –lung function associations have been assessed independently at baseline and at follow up in the said population . It is less surprising that the similar effects were observed for both spirometers since relatively high correlations were observed and indications point that the use of a particular spirometer was not associated with a particular region/subset of the cohort. This would complicate the risk exposure-lung function relationships especially if particular regions have higher PM2.5 levels etc.

Reviewer #2: Thank you for the opportunity to review this valuable sub study from an international multi-centre prospective cohort examining the reliability in measurements between two handheld spirometers. The study uses a large and diverse population to support reliability and generalisability, and presents further findings according to strata and pollution exposure. These findings will support the global health community in ensuring accessibility to lung function testing. I have a number of comments and suggestions for sensitivity and sub analysis that would support the authors’ message and reliability of evidence.

• The authors report little effect between strata according to sex, age category, BMI category, smoking status, and COPD/Asthma however there are no tests reported to confirm lack of significant effect. Either statistical tests or visualisation of data would support the assertions of minimal variation across strata.

• Certain regions have lower percentages of high grade blows and high percentages of low grade blows. Do the authors have any insight into the rationale for differences in grade between regions? The authors could present supplemental sub analysis to investigate the potential source of variability, such as countries within region. There may be more consistent bias within specific countries, which may represent particular factors e.g. socioeconomics.

• Can the authors report the % of blows that are outside the limits of agreement in the bland altman plots/table, and can the authors report any sub analyses that explore the characteristics of participants that were outside these limits compared to within to assess whether there are factors that bias reliability between spirometers. Similarly, it would be interesting to present the grade of EasyOne measures using colours on the bland altmann plots to interpret how grade may have contributed to agreement between spirometers.

• The authors present data from 4,603 participants across 18 countries and 7 regions. The demographics of this population is presented in Table 1. Table 4 is based on data from >100,000 individuals which are not demographically reported, it is not clear whether there is overlap of individuals between MicroGP and EasyOne, and it would be interesting to perform sensitivity analysis in those individuals with data from both spirometers. It would also be possible to then specify spirometer in models as a covariate to see whether there is a significant association.

• The authors could present supplemental information that the first five consecutive participants from each community are representative of the wider PURE study, which would also support Table 4 interpretation.

• Can the authors comment on the possibility of misclassification of COPD and asthma, 7% seems potentially low for COPD and asthma as a single variable. Were numbers representative across all regions, or do they reflect specific ones? In supplement, the authors could present a breakdown of strata by region to assist in interpretation of Table 3.

• The authors should confirm whether assumptions for intraclass coefficients are achieved, such as those regarding normality and variance, as the ICC can be sensitive to this.

• The authors demonstrate good correlation and agreement overall, but can the authors comment on whether studies utilising a mixture of spirometers should apply weightings according to certain (demographic?) factors to support standardisation that may enable more accurate data alignment, in particular whether this valuable dataset can estimate such weightings, or whether the authors are confidence this would not be necessary. Supplementary sensitivity analysis of e.g. grade A-C repeat measures within a spirometer vs between spirometer may give further reliable insight in to comparability of datasets, coefficient of variation may facilitate this.

• There are a few minor typographical errors, for example Table 3 includes 95%CI with some missing or erroneous symbols. E.g. FVC No COPD/Asthma mean Diff % -2.5 (1.6, 3.3): is this a negative difference with erroneous 95%CI or positive difference? Similarly for FEV1 age >65 and 50-65 mean diff %. There may be some issues with 95%CI values for females FVC and males FEV1 that authors should check. Table 4 adjusted model EasyOne-FVC -22,8 rather than -22.8

Reviewer #3: Summary

The authors compared the correlation and agreement between two spirometry devices (MicroGP and EasyOne) in a subset of 4603 participants from the PURE cohort study. Device allocation was random and both tests performed on the same day. Appropriate statistical methods were chosen including Intra-class correlation coefficients and Bland-Altman plots. Results were stratified by clinical and epidemiological characteristics and presented appropriately in figures and tables. Strong correlations were noted between paired FEV1 and FVC measurements, and differences between spirometers were largely within the limits of between effort reproducibility according to ATS/ERS guidelines. Limits of agreement were wide, especially for FVC. Participant characteristics appeared to have minimal impact on agreement between the two devices, however more variation was noted by location especially for South America and the Middle East, which may be clinically significant for FVC. Robust discussion of strengths and limitations was presented. The authors concluded that the use of these two spirometers provide reliable data that are without bias. This conclusion while understandable as the results from the spirometers are highly correlated, is perhaps a little overstretched on account of the wide limits of agreement.

Major concerns

The authors do not provide enough detail on exactly how the random order was generated when deciding which spirometer to use first? This is import to avoid a training effect.

The limits of agreement for FEV1 and FVC are wide and the authors provide justification for this based on previous research in studies with large sample sizes. However, it may be an over stretch to conclude that “Our findings suggest that the use of these two spirometers in large long-term studies provide reliable data that are without bias.” The fact the limits of agreement are wide, suggests some ambiguity in the result especially for FVC, which is likely clinically significant in some of the populations and caution should be taken when forming conclusions.

Minor issues

It would be appropriate in this circumstance to provide a brief description in the methods section as to how the two spirometers work and how they are different. For example, ultrasonic spirometers measure flow independently of gas composition, pressure, temperature, and humidity, and why the two spirometers may be expected to produce different results. This would help give context as to why it is important to compare these two devices.

The authors stated that 65% of participants achieved clinically acceptable measurements using the automated grading criteria generated by the EasyOne device, do they have data from the MicroGP from which you can classify spirometry manoeuvres into these ranges? It is important to know how many participants achieved clinically acceptable results on each device if they are going to quote the 65% value from the EasyOne in the publication. This speaks to the ease of use of the monitors.

It may be useful to include mean percent predicted values for each device (e.g. FEV1, EasyOne 98% of predicted, FEV1 MicroGP 96% of predicted). This will help to clearly display how the difference between devices impacts on the predicted norm for each population.

---

## [Decision Letter · Decision Letter 1]

3 Dec 2021

Differences and agreement between two portable hand-held spirometers across diverse community-based populations in the Prospective Urban Rural Epidemiology (PURE) study.

PGPH-D-21-00083R1

Dear Dr. Duong,

We're pleased to inform you that your manuscript has been judged scientifically suitable for publication and will be formally accepted for publication once it meets all outstanding technical requirements.

Within one week, you'll receive an e-mail detailing the required amendments. When these have been addressed, you'll receive a formal acceptance letter and your manuscript will be scheduled for publication.

An invoice for payment will follow shortly after the formal acceptance. To ensure an efficient process, please log into Editorial Manager at https://www.editorialmanager.com/pgph/ click the 'Update My Information' link at the top of the page, and double check that your user information is up-to-date. If you have any billing related questions, please contact our Author Billing department directly at authorbilling@plos.org.

Kind regards,

Andre F. S. Amaral, Ph.D.

Academic Editor

Additional Editor Comments (optional):

Reviewers' comments:

Reviewer's Responses to Questions

**Comments to the Author**

1. If the authors have adequately addressed your comments raised in a previous round of review and you feel that this manuscript is now acceptable for publication, you may indicate that here to bypass the “Comments to the Author” section, enter your conflict of interest statement in the “Confidential to Editor” section, and submit your "Accept" recommendation.

Reviewer #2: All comments have been addressed

Reviewer #3: All comments have been addressed

2. Does this manuscript meet PLOS Global Public Health’s publication criteria? Is the manuscript technically sound, and do the data support the conclusions? The manuscript must describe methodologically and ethically rigorous research with conclusions that are appropriately drawn based on the data presented.

Reviewer #2: Yes

Reviewer #3: Yes

3. Has the statistical analysis been performed appropriately and rigorously?

Reviewer #2: Yes

Reviewer #3: Yes

4. Have the authors made all data underlying the findings in their manuscript fully available (please refer to the Data Availability Statement at the start of the manuscript PDF file)?

Reviewer #2: Yes

Reviewer #3: Yes

5. Is the manuscript presented in an intelligible fashion and written in standard English?

Reviewer #2: Yes

Reviewer #3: Yes

6. Review Comments to the Author

Reviewer #2: Thank you to the authors for comprehensively addressing the review comments by undertaking major revisions to the presented findings and methodology. I note that the interpretations are reliable and the reported findings offer added insights regarding the sources of variability. Additionally, I note a number of tables and figures have been provided in appended results which offer further focus and justification of author interpretation. I have no further comments.

Reviewer #3: Thank you for the opportunity to review this resubmission. The authors appear to have made considerable effort to address to comments made at earlier review. Although not completely explanatory, they have defined that the randomisation procedure to generate the order of spirometry use was done centrally by the coordinating centre. By incorporating the GLI equations into their study, they have demonstrated that the z-score differences between spirometers was small and within an acceptable range so not to impact interpretation. In addition, the wording in the conclusions section has been adjusted so not to overstate their findings. Authors have provided a clear explanation of how the two spirometers work; this adds context to their paper as to why a comparative study is necessary. They have also provided interesting data which explores where the variation between spirometers originates, unsurprisingly the majority of variation was identified at the individual level. Overall, the changes have much improved the paper, which will be a valuable resource for other longitudinal studies where it is necessary to change spirometers.

7. PLOS authors have the option to publish the peer review history of their article (what does this mean?). If published, this will include your full peer review and any attached files.

**Do you want your identity to be public for this peer review?** For information about this choice, including consent withdrawal, please see our Privacy Policy.

Reviewer #2: **Yes: **Dr Iain Stewart

Reviewer #3: **Yes: **Ben Knox-Brown
